# Gestational weight gain and its effect on birth outcomes in sub-Saharan Africa: Systematic review and meta-analysis

**Fekede Asefa**[1,2,3]*, **Allison Cummins**[2], **Yadeta Dessie**[1], **Andrew Hayen**[3], **Maralyn Foureur**[2,4]

**1** School of Public Health, College of Health and Medical Sciences, Haramaya University, Harar, Ethiopia, **2** Centre for Midwifery, Child and Family Health, Faculty of Health, University of Technology Sydney, Ultimo, NSW, Australia, **3** Australian Centre for Public and Population Health Research, Faculty of Health, University of Technology Sydney, Ultimo, NSW, Australia, **4** Hunter New England Health, Nursing and Midwifery Research Centre, University of Newcastle, Callaghan, NSW, Australia

* sinboona@gmail.com

## Abstract

### Introduction

An increased metabolic demand during pregnancy is fulfilled by gaining sufficient gestational weight. Women who gain inadequate-weight are at a high-risk of premature birth or having a baby with low-birth weight. However, women who gain excessive-weight are at a high-risk of having a baby with macrosomia. The aim of this review was to determine the distribution of gestational weight gain and its association with birth-outcomes in Sub-Saharan Africa.

### Methods

For this systematic review and meta-analysis, we performed a literature search using PubMed, Medline, Embase, Scopus, and the Cumulative Index to Nursing and Allied Health Literature (CINAHL) databases. We searched grey-literature from Google and Google Scholar, and region-specific journals from the African Journals Online (AJOL) database. We critically appraised the included studies using the Effective Public Health Practice Project Quality Assessment Tool for Quantitative Studies. Two independent reviewers evaluated the quality of the studies and extracted the data. We calculated pooled relative-risks (RR) with 95% confidence intervals.

### Results

Of 1450 retrieved studies, 26 met the inclusion criteria. Sixteen studies classified gestational weight gain according to the United States Institute of Medicine recommendations. The percentage adequate amount of gestational weight ranged from 3% to 62%. The percentage of inadequate weight was >50% among nine studies. Among underweight women, the percentage of women who gained inadequate gestational weight ranged from 67% to 98%. Only two studies were included in the meta-analyses to evaluate the association of

**Data Availability Statement:** All relevant data are within the manuscript and its Supporting Information files.

**Funding:** The author(s) received no specific funding for this work.

**Competing interests:** The authors have declared that no competing interests exist.

**Abbreviations:** AJOL, African Journal Online; ANC, Antenatal Care; BMI, Body Mass Index; CI, Confidence Interval; GWG, Gestational Weight Gain; IOM, Institute of Medicine; LBW, Low Birth Weight; PRISM-P, Preferred Reporting Items for Systematic Review and Meta-Analysis Protocols; RR, Relative Risk; SSA, sub-Saharan Africa; WHO, World Health Organization.

gestational weight gain with pre-eclampsia and macrosomia. No difference was observed among women who gained inadequate and adequate gestational weight regarding experiencing pre-eclampsia (RR, 0.71; 95% CI: 0.22, 2.28, P = 0.57). Excessive gestational weight gain was not significantly associated with macrosomia compared to adequate weight gain (RR, 0.68; 95% CI: 0.38, 1.22, P = 0.20).

## Conclusion

A substantial proportion of sub-Saharan African women gain inadequate gestational weight particularly high among underweight women. Future interventions would need to design effective pre-pregnancy weight management strategies.

## Introduction

Desirable gestational weight gain (GWG) supports the increased metabolic demands required for positive pregnancy outcomes [1]. Guidelines regarding appropriate levels of GWG have been promoted worldwide [2]. A variety of guidelines about the GWG exist; their approach in GWG management also varies [3, 4]. More than a half (54%) of the GWG guidelines are similar to the 2009 United State Institutes of Medicine (IOM) recommendations [3]. The IOM recommends that women gain between 0.5 and 2.0 kg in the first trimester of pregnancy. In the second and third trimester, the weight gain recommendation is 0.44 to 0.58 kg per week for women who were underweight during the pre-pregnancy period (body mass index (BMI) $\leq$18.5 kg/m$^2$); 0.35 to 0.50 kg per week for women of normal-weight (BMI is 18.5 to <25 kg/m$^2$); 0.23 to 0.33 kg per week for over-weight women (BMI 25 to 30 kg/m$^2$); and 0.17 to 0.27 kg per week for obese women (BMI $\geq$30 kg/m$^2$). In total, the IOM recommends weight gain of 12.5 to 18 kg for underweight women; 11.5 to 16 kg for normal weight women, 7 to 11 kg for overweight women and 5 to 9 kg for obese women [5].

The amount of weight gained during pregnancy is determined by factors including the mother's age [2, 6, 7], parity [2, 6, 7], income status [2, 8], educational status [7], social class [6], and pre-pregnancy maternal weight [2, 9]. Other factors include antenatal care (ANC) [2, 6], physical activity during pregnancy [10] and perinatal depression[11–13].

A desirable GWG is essential for optimal outcomes for both the mother and her baby [14]. Inappropriate GWG can pose health risks for mother and baby [15, 16]. Women who do not gain enough weight during pregnancy are at risk of having a baby with low birth weight (LBW) [17, 18] and pre-term birth [19]. Women who gain excessive weight are at an increased risk of hypertension in pregnancy, as well as an increased risk of pre-eclampsia [20–22], gestational diabetes [15, 20], caesarean sections [20, 22, 23], postpartum haemorrhage [22], postpartum weight retention [24], and development of long-term obesity [25].

The World Health Organization (WHO) defines low birth weight as a birth weight less than 2500g [26]. Globally, LBW contributes to 60% to 80% of all neonatal deaths [27]. About 95% of the 20.6 million LBW infants born each year are in low-income countries [26, 27]. Inadequate GWG in combination with low pre-pregnancy weight is associated with higher rates of LBW and prematurity [28].

To date, there are few systematic reviews and meta-analyses of research in sub-Saharan Africa (SSA) on the weight of pregnant women [29–31]. None addressed how much weight is gained during pregnancy by women in this population, or the effect on birth outcomes. Therefore, this systematic review and meta-analysis presents available evidence on the amount of

GWG, factors affecting GWG and the association of GWG with birth outcomes, in sub-Saharan Africa.

## Methods

### The protocol and registration

The method of this systematic review and meta-analysis was reported using the Preferred Reporting Items for Systematic Review and Meta-Analysis Protocols (PRISMA-P) 2015 statement recommendations [32] (S1 Table). We followed the flowchart from the PRISMA-P 2015 guideline recommendation to demonstrate the selection process from initially identified records to finally included studies [33]. The protocol for this review was registered on the International Prospective Register of Systematic Reviews (PROSPERO) registration number CRD42018085499 [34].

### Search strategy

We identified literature from PubMed, Medline, Embase, Scopus, and CINAHL databases (S2 Table). We also accessed the African Journals Online (AJOL) database for papers published in country-specific or region-specific journals. A supplementary search was conducted to find grey literature from the Google search engine and Google Scholar. In addition, we contacted six authors to request additional information missing from their papers. However, only one author [35] responded to the email request. The search was limited to papers published since 1990 (when the IOM guideline was published [36]) to 2019 in sub-Saharan Africa and published in English. We employed the Medical Subject Headings (MeSH) terms, Emtree, CINAHL headings and combined keywords to identify studies in these databases. The search terms emerged from the following keywords (GWG, Weight gain during pregnancy, Birth outcome, Birth weight, Low birth weight, sub-Saharan Africa).

### Eligibility criteria

We included cross-sectional, case-control, cohort and randomized controlled trials. We included studies that defined GWG as inadequate, adequate, or excess according to IOM recommendations, or mean GWG in total or in each trimester, and that explicitly reported for underweight, normal weight, overweight and obese women (based on pre-pregnancy BMI). We also included studies that classified GWG based on the researchers' categories and studies that assessed the association of GWG with birth outcomes. We excluded the studies if they were duplicates; anonymous reports; not published in English language; systematic reviews and meta-analyses or studies that were unable to provide information about the adequacy of GWG. The primary outcome of interest in this study is GWG. Other outcomes were factors affecting GWG and the association between GWG and birth outcomes.

### Study selection procedure

We located an initial set of studies by using the search terms and applying filters to the databases. We exported the identified studies to Covidence, a systematic review software [37], and we excluded duplicates. Two reviewers independently screened the studies based on titles and abstracts as per the inclusion criteria. During the screening process, we resolved any disagreements between the two reviewers through discussion. However, in the case of further disagreement, other authors made the final decisions.

## Quality assessment

Two independent appraisers appraised the quality of the included studies. We used the Effective Public Health Practice Project Quality Assessment Tool for Quantitative Studies [38] is to appraise the studies critically and to report the level of the strength of a study's quality. The quality assessment tool uses a number of criteria to rate the strength of the studies. These criteria include the presence of selection bias, the strength of the study design, withdrawals and dropout rate, data collection practices, blinding as part of a controlled trial and how confounders were controlled. Each examined practice paper marked as "strong," "moderate," or "weak". During appraisal, attention was given to the clear description of objectives, inclusion criteria, precision of measurement of the outcome (the time and how pre-pregnancy BMI and GWG were measured) and the appropriateness of statistical analyses.

## Data extraction process

We used an excel spreadsheet for data extraction. Two reviewers extracted the data using a data extraction format which includes authors, year of publication, study design, sample size, the country of the study, objectives of the study, how GWG was measured, time at which pre-pregnancy BMI was measured, and the pre-pregnancy weight status of the women (underweight, normal-weight, overweight, and obese). We extracted data on GWG (mean for each category of pre-pregnancy weight, the percentage of inadequate, adequate or excess), factors affecting GWG and effects of GWG on birth-outcomes. Where the GWG categorisation did not follow the IOM categories, we used the categorisation used in the study.

## Data analysis

Findings from each study were described by the country of the studies, population characteristics, women's pre-pregnancy BMI, study design, study objectives, and outcomes. Outcomes, GWG, were reported using the IOM classification. For studies that used arbitrary classifications (for example, $\leq$ 8.0 kg (inadequate GWG), 8.1 to 16.0 kg (adequate GWG), and $\geq$16.1 kg (excessive GWG) [39]; or <7 kg (inadequate GWG), 7 to 12 kg (adequate t GWG), and >12 kg (excessive GWG) [40]), we used the authors' own classifications.

We used forest plots to report the results graphically. We checked the presence of heterogeneity among studies using the chi-squared test where statistical significance with a p-value <0.05. The $I^2$ statistic was used to quantify the level of heterogeneity among the studies. We assumed substantial heterogeneity among studies when the value of $I^2$ was $\geq$50%. We used the Mantel–Haenszel fixed effects model to conduct meta-analyses where the studies did not have substantial heterogeneity (i.e. $I^2$ statistic < 50%). We used random effects model while assessing the effect of gaining inadequate gestational weight on pre-eclampsia although the $I^2$ value is <50%, because we have observed considerable heterogeneity among included studies. We pooled the percentages of inadequate, adequate and excess GWG. However, substantial heterogeneity was detected among studies ($I^2$ value for inadequate, adequate and excess weight gain were 99.7%, 98.9% and 99.1% respectively) (S1 to S3 Figs). We stratified women into underweight, normal-weight, overweight and obese women to pool their GWG, but the $I^2$ value within each group of the women was >95.0% (S4 to S6 Figs). The association between GWG and birth outcomes (LBW, Macrosomia, APGAR-score, caesarean section, obstetric hemorrhage, pre-eclampsia, and episiotomy) was determined using the Review Manager Software (RevMan version 5.3 for windows) [41]. We calculated risk ratios with 95% confidence intervals. However, due to high heterogeneity among studies and the limited number of studies (S3 and S4 Tables), we reported only the association between GWG and macrosomia and pre-

eclampsia. Factors associated with GWG were classified differently among different studies. We used narrative synthesis to describe factors associated with GWG.

# Results

## Results of the screening process

The search retrieved a total of 1450 studies. A total of 1086 articles were reviewed after removal of 364 duplicates. Based on title and abstract screening, we excluded 964 articles, and we conducted a full-text review on the remaining 121 studies left. We included 26 studies in the review. The most common reasons for exclusion were failure to report GWG according to IOM recommendations or failure to explicitly report pre-pregnancy weight specific GWG or only reporting weight gain that did not indicate the adequateness of GWG (Fig 1).

## Study characteristics

Table 1 describes the characteristics of the studies included in this review. Five studies were from Nigeria [6, 7, 42–44]; four from Cameroon [45–48]; four each from Ethiopia [2, 35, 49, 50] and Ghana [39, 51–53]; two studies from South Africa [54, 55] and Malawi [28, 56]; and one each from Uganda [9], Kenya [57], Niger [58], Benin [40], and the Democratic Republic of Congo [59]. Based on a country's income status [60], two studies were from upper middle-income countries [54, 55]; fourteen from lower middle-income countries [6, 7, 39, 42–48, 51–53, 57] and ten from low-income countries [2, 9, 28, 35, 40, 49, 50, 56, 58, 59].

Sixteen studies [2, 7, 9, 28, 35, 42, 45, 47, 51–56, 58, 59] classified the outcome (GWG) according to the IOM recommendations, but for seven studies [6, 39, 40, 43, 44, 48, 50] standard criteria were not used to measure and classify the outcome, that is the authors classified weight gain using their own method. Three studies reported according to the International Fetal Newborn Growth Standards for the 21st Century (INTERGROWTH-21st) guidelines [52, 57, 58], of which two studies [52, 58] used both IOM 2009 and INTERGROWTH-21st guidelines. The authors of one study stated that they used the IOM classification, but they also reported normal weight gain as "women with BMIs between 18.5 kg/m² and 30 kg/m² and who gained 9 to 16 kg; excessive weight gain for those who gained weight above these ranges" [46]. Eight studies reported GWG separately for each category of woman's pre-pregnancy weight [2, 7, 9, 35, 42, 53, 54, 56], and the author of one study provided these data upon email request [35]. Four studies [9, 44, 45, 48] used self-reported pre-pregnancy weight while three studies [52, 54, 59] used weight after 20 weeks of gestation and one study [56] used weight at 24 weeks of gestation. It was not clear when and how pre-pregnancy weight was measured in two studies [46, 58] (Table 1).

## Critical appraisal results

In two studies, loss to follow-up was not well described. In one of these studies [7], a cohort of 1000 women was recruited, but the authors reported the results of 590 women, but there was not an adequate description of loss to follow-up of the remaining 410 women. Poor control of confounding factors was also an issue affecting the quality of the studies [6, 9, 39, 42, 43, 46–48]. These studies either did not control for confounding factors at all or did not include all necessary variables into the analysis (partially controlled) or did not report how confounding was controlled. According to our quality assessment, 17 studies had moderate quality, while the remaining 10 studies had weak quality. Except for one study [52], all included studies were observational studies (Table 2).

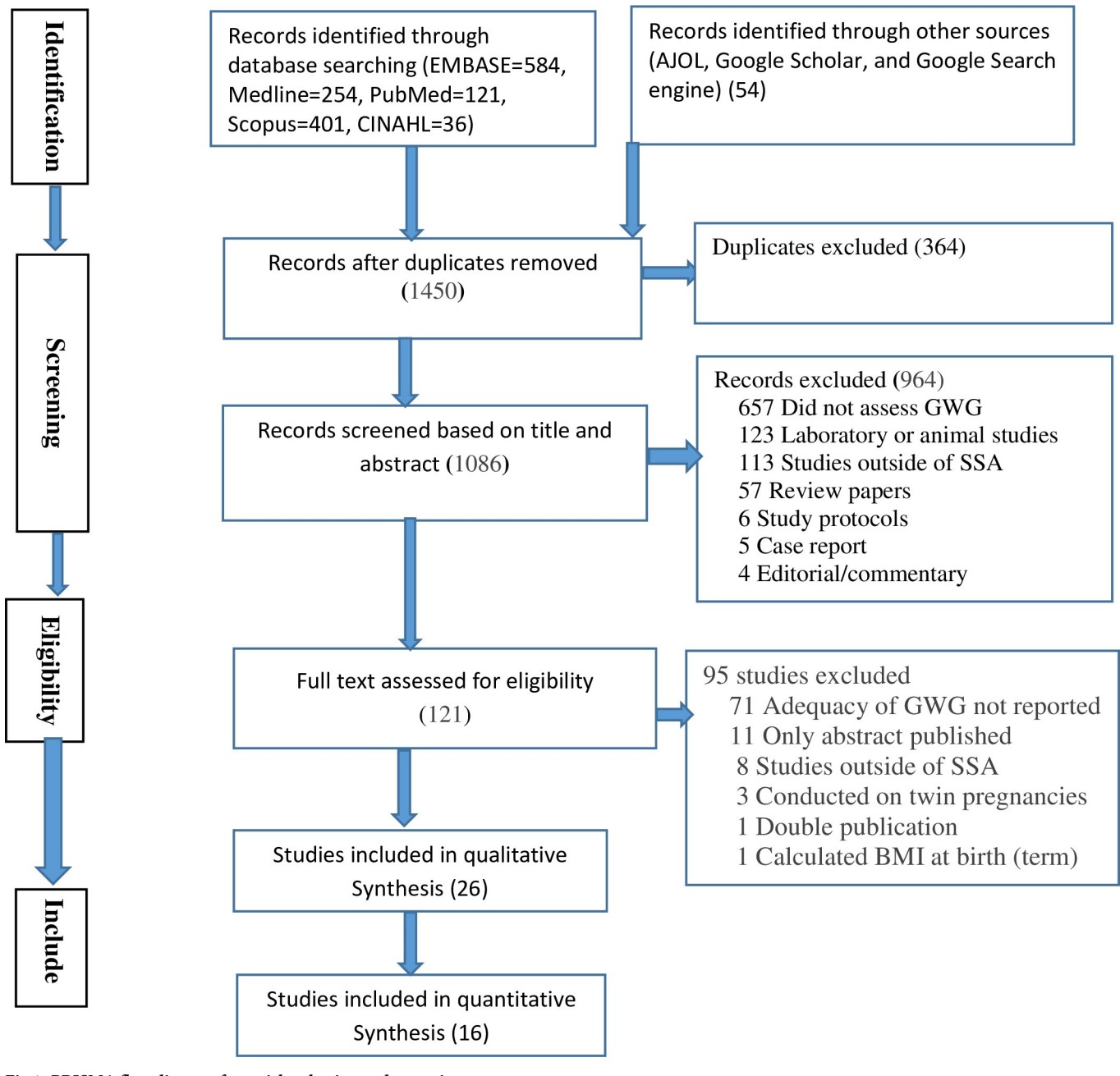

**Fig 1. PRISMA flow diagram for article selection and screening.**

## Gestational weight gain classifications

**Gestational weight gain according to IOM classification.** Sixteen studies reported the percentage of GWG according to IOM recommendations. The percentage of women with inadequate GWG ranged from 15.7% to 96.6% [7, 55]. The percentage of women with adequate GWG ranged from 3% to 62% [7, 42]. Nine of the 16 studies reported the percentage of women with inadequate GWG as >50% [2, 7, 9, 28, 35, 52, 56, 58, 59] and the percentage of women with adequate GWG as <30% [2, 7, 28, 35, 52–54, 56, 58, 59]. The smallest percentage of inadequate GWG (15.7%)[55] and the highest percentage of excessive GWG (55.5%) [54]

Table 1. Characteristics of the 21 studies reporting on gestational weight gain in relation to pre-pregnancy weight in sub-Saharan Africa, 2019.

| Author and year | Country | Study design | Study settings | Sample size | Objective of the study | GWG Measure | Time at which pre-pregnancy BMI measured | Underweight (UW), normal-weight (NW), Overweight(OW and Obese (O) women Number (%) |
|---|---|---|---|---|---|---|---|---|
| Fouelifack FY et al 2015 [45] | Cameroon | Retrospective Cohort | Urban Referral hospital | 465 | To assess associations of BMI and GWG with pregnancy outcomes | IOM 2009 | Self-reported pre-pregnancy weight | UW = 17 (3.7) NW = 228(49) OW = 152(32.7) O = 65(14) |
| Mbu RE et al 2013 [46]* | Cameroon | Cross-sectional study | Urban Maternity hospital (type of the hospital is indicated) | 220 | To determine pregnancy outcomes among women who gained normal and excess gestational weight | IOM 2009 with modifications | Not clearly stated | Not clearly stated |
| Asefa F et al 2016 [2] | Ethiopia | Cross-sectional study | Urban Both primary and referral hospitals | 411 | To assess GWG and associated factors | IOM 2009 | Before 16 weeks of gestation | UW = 39 (9.5) NW = 296(72) OW = 60(14.6) O = 16(3.9) |
| Halle-Ekane GE et al 2015 [47] | Cameroon | Cross-sectional study | Urban District hospitals | 350 | To determine the prevalence of excessive GWG, its risk factors, and effects on pregnancy outcomes | IOM 2009 | Before 13 weeks of gestation | UW = 8 (2.3) NW = 176 (50.3) OW = 115(32.8) O = 51(14.6) |
| Seifu B 2017 [35] | Ethiopia | Cross-sectional study | Urban Health centre, primary and referral hospitals | 549 | To compare GWG and its associated factors among HIV-positive and HIV-negative women | IOM 2009 | Before 16 weeks of gestation | UW = 107 (19.5) NW = 371 (67.6) OW = 65 (11.8) O = 6 (1.1) |
| Abubakari A et al 2015 [51] | Ghana | Cross-sectional study | Urban, peri-urban and rural Both primary and referral hospitals | 419 | To assess the association between pre-pregnancy BMI, GWG, maternal socio economic and demographic factors and birth weight | IOM 2009 | First trimester | UW = 16 (3.8) NW = 242 (57.8) OW = 105 (25.0) O = 56 (13.3) |
| Wanyama R et al. 2016 [9] | Uganda | Cross-sectional study | Urban Health centre | 192 | To determining the prevalence of inadequate, adequate and excessive GWG | IOM 2009 | Self-reported pre-pregnancy weight | UW = 28 (14.6) NW = 143 (74.5) OW = 21 (10.9) O = 0 |
| Wrottesley SV et al 2017 [54] | South Africa | Prospective cohort study | Urban Teaching hospitals | 538 | To assess patterns of habitual dietary intake and their associations with first trimester BMI and GWG | IOM 2009 | Before 20 weeks of gestation | UW = 0 NW = 182 (33.8) OW = 190 35.3) O = 166 (30.9) |
| Chithambo SET 2017 [56] | Malawi | Longitudinal study (Cohort) | Rural Community based | 257 | To identify factors associated with the rate of GWG | IOM 2009 | Before 24 weeks of gestation | UW = 18 (7.0) NW = 201 (78.2) OW = 38 (14.8) O = 0 |
| Esimai OA et al 2014 [7] | Nigeria | Longitudinal (cohort)study | Urban vs rural is not clearly stated Primary health facilities | 590 | To determine correlates of gestational weight gain and infant birth weight | IOM 2009 with some operational definition (<7 kg low, >7 kg high) | First 2 months of pregnancy | UW = 47 (8.0) NW = 482 (81.7) OW = 46 (7.8) O = 15 (2.5) |
| Iyoke CA et al 2013 [42] | Nigeria | Retrospective cohort | Urban Teaching hospitals | 648 | To compare GWG and obstetric outcomes between obese and normal weight women | IOM 2009 | First trimester | UW = NA NW = 324 (50.0) OW = NA O = 324 (50.0) |

(Continued)

**Table 1.** (*Continued*)

| Author and year | Country | Study design | Study settings | Sample size | Objective of the study | GWG Measure | Time at which pre-pregnancy BMI measured | Underweight (UW), normal-weight (NW), Overweight(OW and Obese (O) women Number (%) |
|---|---|---|---|---|---|---|---|---|
| Adu-Afarwuah S et al 2017 [52] | Ghana | Randomized Controlled Trial | Semi-urban Primary hospitals and poly clinic | 1320 | To determine the association of SQ-LNSs with differences in GWG or maternal anthropometric characteristics, including risk of overweight or obesity | IOM 2009 and INTERGROWTH-21st guidelines | Before 20 weeks of gestation | UW = 50 (3.8) NW = 743 (56.3) OW = 354 (26.8) O = 143 (10.8) |
| Nomomsa D et al 2014 [49] | Ethiopia | Cross-sectional study | Urban Both primary and referral hospitals | 411 | To assess the association of GWG and LBW | IOM 2009 | Before 16 weeks of gestation | UW = 39 (9.5) NW = 296(72) OW = 60(14.6) O = 16(3.9) |
| Muyayalo KP et al 2017 [59] | Democratic Republic of Congo | Prospective Cohort | Urban Referral hospitals | 199 | To determine proportion of post-partum weight retention and its average level; to identify its risk factors; to determine the proportion of obese women 6 weeks after delivery. | IOM 2009 | Before 20 weeks of gestation | UW = 11 (5.5) NW = 111 (55.8) OW = 56 (28.1) O = 21 (10.6) |
| Ismail LC et al 2016 [57] | Kenya | longitudinal (cohort) study | Urban (institution where the study collected was not clearly indicated) | Varies[§] | To describe patterns in maternal gestational weight gain in healthy pregnancies with good maternal and perinatal outcomes | Mean GWG at each month of follow-up and INTERGROWTH-21st[†] | Before 14 weeks of gestation | All were normal weight women |
| Addo VN 2010 [39] | Ghana | Cross-Sectional study | Urban vs rural is not clearly stated Private specialist Hospital | 1755 | To find out the effects of pregnancy weight gain in different BMI groups on maternal and neonatal outcomes | Operationally defined (Low weight gain ≤ 8.0 kg, Normal weight gain 8.1 to 16.0 kg, High weight gain ≥16.1 kg) | Between 10 and 13 weeks of Gestation | UW = 77 (4.4) NW = 832 (47.4) OW = 609 (34.7) O = 314 (17.9)[‡] |
| Onwuka CI et al. 2017 [6] | Nigeria | Longitudinal (cohort) study | Urban Teaching hospitals | 200 | To determine the pattern of GWG and its association with birth weight | Operationally defined (<10 kg inadequate, 10 to 15 kg adequate, >15 kg excess) | Before 14 weeks of gestation | UW = 7 (3.5) NW = 102 (51.0) OW = 35 (17.5) O = 56 (28.0) |
| Elie N et al 2015 [48] | Cameroon | Cross-sectional study | Urban University teaching hospital | 232 | To identify risk factors for a baby born with macrosomia | Operationally defined (<16 kg and ≥16 kg) | Before 20 weeks of gestation but from maternal recall before she realized pregnancy | UW = 0 NW = 114 (49.1) OW & O = 118 (50.9) |
| Onyiriuka A. N 2006 [43] | Nigeria | Cross-sectional study | Urban Referral Hospital | 408 | To determine the incidence of delivery of HBW (macrosomia) | Operationally defined (<10 kg, 10 to 12 kg, 13 to 15 and >15 kg) | First trimester | Not reported |

(*Continued*)

**Table 1.** (Continued)

| Author and year | Country | Study design | Study settings | Sample size | Objective of the study | GWG Measure | Time at which pre-pregnancy BMI measured | Underweight (UW), normal-weight (NW), Overweight(OW and Obese (O) women Number (%) |
|---|---|---|---|---|---|---|---|---|
| Akindele RL et al 2017 [44] | Nigeria | Case–control study | Urban Major public hospitals (type of the hospitals are not indicated) | 240 | To determine the incidence of macrocosmic new-borns, their maternal socio-biologic predictors, the neonatal complications attributable to the mode of delivery, and their early neonatal outcome | Operationally defined (<15 kg and ≥15 kg) | Self-reported pre-pregnancy weight | UW, NW & OW = 172 (71.7) O = 68 (28.3) |
| Ward E et al 2007 [55] | South Africa | Longitudinal (cohort) study | Urban vs rural is not clearly stated Primary health care clinic | 89 | To evaluate the association between pre-pregnancy BMI and maternal pregnancy weight gain and pregnancy outcome | IOM 1990 | 14 weeks of Gestation | UW = 14 (15.7) NW = 45 (50.6) OW & O = 28 (31.5) |
| Ouédraogo CT et al 2019 [58] | Niger | Cross-sectional study | Community-based survey | 1386 | To estimate the prevalence and the determinants of low GWG and low mid-upper arm circumference | IOM 2009 and INTERGROWTH-21st guidelines | Not clear (women included regardless of their gestational age) | Not reported |
| Gondwe A et al 2018 [28] | Malawi | Retrospective cohort nested with randomized controlled trial | Semi-urban and semi-rural Private hospital and public health centre | 1287 | To examined whether maternal pre-pregnancy BMI and GWG are associated with birth outcomes | IOM 2009 | Before 20 weeks of gestation | UW = 76 (5.9) NW = 1071 (83.2) OW & O = 140 (10.9) |
| Agbayizah DE 2017 [53] | Ghana | Cross-sectional study | Urban Semi-urban Rural General Hospital | 322 | To assess the prevalence of inadequate, adequate and excessive GWG and its associated factors | IOM 2009 | Before 20 weeks of gestation | UW = 3 (1.0) NW = 164 (50.9) OW = 119 (36.9) O = 56 (11.2) |
| Agbota G et al 2019 [40] | Benin | Longitudinal (cohort) study | Semi-urban and Rural; institution where the study collected was not clearly indicated | 260 | To assess the effect of maternal anthropometric status before conception and during pregnancy on fetal and postnatal growth, up to 12 months of age | Operationally defined (<7 kg, 7 to 12 kg and >12 kg) | Before 7 weeks of gestation | UW = 23 (8.9) NW = 175 (67.3) OW = 43 (16.5) O = 19 (7.3) |
| Tela FG et al 2019 [50] | Ethiopia | Cross-sectional study | Urban Private clinics | 309 | To determine the prevalence of macrosomia and investigate the associated risk factors | Operationally defined (<16 kg, ≥16 kg) | Around 12 weeks of gestation | UW = 28 (9.0) NW = 173 (56.0) OW = 76 (24.6) O = 32(10.4) |

*intentionally included equal number of women who gained excessive gestational weight and who gained adequate gestational weight to compare their birth outcomes

†International Fetal Newborn Growth Standards for the 21st Century -INTERGROWTH-21st (This study is a multicenter study including one sub-Saharan African country, Kenya. However, the GWG according to the INTERGROWTH-21st standard was not explicitly reported for Kenya)

‡ the summation of the described numbers of UW, NW, OW and O women is greater than the described total sample size.

§Varies across Gestational ages (355 for 14–18+6 weeks, 356 for 19–23+6 weeks, 360 for 24–28+6 weeks, 355 for 29–33+6 weeks, 388 for 34–40+0 weeks)

**Table 2. Summary of the quality of included studies according to the Effective Public Health Practice Project Quality Assessment Tool for Quantitative Studies, 2019.**

| Author and year | Selection bias | Study design | Confounder | Blinding | Data collection method | Withdrawal and dropout | Overall strength |
|---|---|---|---|---|---|---|---|
| Fouelifack FY et al 2015 [45] | Moderate | Weak | Moderate | NA* | Moderate | Strong | Moderate |
| Mbu RE et al 2013 [46] | Weak | Weak | Weak | NA | Moderate | Weak | Weak |
| Asefa F et al 2016 [2] | Moderate | Weak | Strong | NA | Moderate | Strong | Moderate |
| Halle-Ekane GE et al 2015 [47] | Strong | Weak | Weak | NA | Strong | Strong | Weak |
| Seifu B 2017 [35] | Moderate | Weak | Moderate | NA | Moderate | Strong | Moderate |
| Abubakari A et al 2015 [51] | Strong | Weak | Strong | NA | Strong | Strong | Moderate |
| Wanyama R et al. 2016 [9] | Moderate | Weak | Weak | NA | Strong | Strong | Weak |
| Wrottesley SV et al 2017 [54] | Strong | Weak | Strong | NA | Strong | Strong | Moderate |
| Chithambo SET 2017 [56] | Strong | Weak | Strong | NA | Strong | Strong | Moderate |
| Esimai OA et al 2014 [7] | Moderate | Weak | Moderate | NA | Strong | Weak | Weak |
| Iyoke CA et al 2013 [42] | Moderate | Weak | Weak | NA | Strong | Strong | Weak |
| Adu-Afarwuah S et al 2017 [52] | Strong | Strong | Strong | Weak | Strong | Strong | Moderate |
| Nemomsa D et al 2014 [49] | Moderate | Weak | Strong | NA | Moderate | Strong | Moderate |
| Muyayalo KP et al 2017 [59] | Moderate | Weak | Strong | NA | Moderate | Strong | Moderate |
| Ismail LC et al 2016 [57] | Strong | Weak | Strong | NA | Strong | Strong | Moderate |
| Addo VN 2010 [39] | Moderate | Weak | Weak | NA | Moderate | Strong | Weak |
| Onwuka CI et al. 2017 [6] | Moderate | Weak | Weak | NA | Strong | Strong | Weak |
| Elie N et al 2015 [48] | Strong | Weak | Weak | NA | Strong | Strong | Weak |
| Onyiriuka A.N 2006 [43] | Strong | Weak | Weak | NA | Strong | Strong | Weak |
| Akindele RL et al 2017 [44] | Moderate | Weak | Strong | NA | Strong | Strong | Moderate |
| Ward E et al 2007 [55] | Moderate | Weak | Moderate | NA | Moderate | Strong | Moderate |
| Ouédraogo CT et al 2019 [58] | Strong | Weak | Strong | NA | Strong | Strong | Moderate |
| Gondwe A et al 2018 [28] | Strong | Weak | Strong | NA | Strong | Strong | Moderate |
| Agbayizah DE 2017 [53] | Moderate | Weak | Moderate | NA | Strong | Strong | Moderate |
| Agbota G et al 2019 [40] | Strong | Weak | Strong | NA | Strong | Strong | Moderate |
| Tela FG et al 2019 [50] | Moderate | Weak | Moderate | NA | Moderate | Strong | Moderate |

*Not-applicable

were from South Africa. In 11 of the 16 studies, the percentage of women with excessive GWG was <20% [2, 7, 9, 28, 35, 42, 51, 52, 56, 58, 59] (Table 3).

Of the eight studies [2, 7, 9, 35, 42, 53, 54, 56] that reported GWG separately for each category of women's pre-pregnancy weight, two studies had no underweight women [42, 54] or obese women [9, 56], while one study had no overweight women [42]. According to the six studies that had underweight women [2, 7, 9, 35, 53, 56], more than 67% of underweight women were reported to have gained inadequate gestational weight. In four studies [2, 7, 9, 56], more than two-thirds of normal weight women gained inadequate gestational weight, but in three studies [42, 53, 54], nearly one -third of normal weight women gained inadequate gestational weight. As pre-pregnancy BMI of the women increased, the percentage of those with adequate GWG increased (7.7% among underweight women and 62.5% among obese women [2]; 2.1% among underweight women and 93.3% among obese women [7]) (Table 4).

**Mean gestational weight gain.** According to one study, mean GWG (± standard deviation) was 1.52±1.65 kg during 14 to 18$^{+6}$ weeks; 2.57±1.46kg during 19 to 23$^{+6}$ weeks; and 2.48 ±1.29 kg during 24 to 28$^{+6}$ weeks. Similarly, GWG during 29 to 33$^{+6}$ weeks, and 34 to 40$^{+0}$ weeks was 2.18 ±1.39 kg and 2.42±2.41 kg, respectively [57]. According to the study from

**Table 3. Studies describing proportions of inadequate, adequate and excess gestational weight gain in Sub-Saharan Africa according to the United State Institute of Medicine recommendations, 2019.**

| Authors and year | Sample Size | Inadequate GWG n (%) | Adequate GWG n (%) | Excess GWG n (%) |
|---|---|---|---|---|
| Chithambo SET et al. 2017 [56] | 257 | 206 (80.2) | 51 (19.8) | 0 (0.0) |
| Asefa F et al. 2016 [2] | 411 | 285 (69.3) | 115 (28.0) | 11 (2.7) |
| Seifu B et al. 2017 [35] | 549 | 369 (67.2) | 160 (29.2) | 20 (3.6) |
| Wanyama R et al. 2016 [9] | 192 | 120 (62.5) | 66 (34.4) | 6 (3.1) |
| Esimai OA et al 2014 [7] | 590 | 570 (96.6) | 18 (3.1) | 2 (0.3) |
| Abubakari A et al 2015 [51] | 419 | 208 (49.6) | 180 (43.0) | 31 (7.4) |
| Adu-Afarwuah S et al 2017 [52] | 1030 | 646 (62.7) | 277 (26.9) | 107 (10.4) |
| Muyayalo K P et al 2017 [59] | 199 | 117 (58.8) | 52 (26.1) | 30 (15.1) |
| Iyoke CA et al 2013 [42] | 648 | 121 (18.7) | 400 (61.7) | 127 (19.6) |
| Halle-Ekane GE et al 2015 [47] | 350 | 129 (36.9) | 114 (32.6) | 107 (30.6) |
| Fouelifack FY et al 2015 [45] | 462 | 131 (28.0) | 186 (40.0) | 145 (32.0) |
| Wrottesley SV et al 2017 [54] | 538 | 128 (24.0) | 113 (21.0) | 297 (55.5) |
| Ward E et al 2007 [55] | 89 | 14 (15.7) | 46 (51.7) | 29 (29.6) |
| Ouédraogo CT et al 2019 [58] | 911 | 574 (63.0) | 218 (24.0) | 119 (13.0) |
| Gondwe A et al 2018 [28] | 1287 | 924 (71.8) | 296 (23.0) | 67 (5.2) |
| Agbayizah ED 2017 [53] | 322 | 73 (22.7) | 94 (29.2) | 155 (48.1) |

Uganda weekly mean GWG of 0.32 kg, 0.30 kg and 0.28 kg were reported among underweight, normal-weight and overweight women, respectively [9]. In another study, the mean GWG was 9.14±3.46 among underweight women; 9.26±3.14 kg among normal-weight women; 8.03±3.64 kg among overweight women, and 6.44±3.46 kg among obese-women [2]. Onwuka et al also reported a mean GWG of 10.21±2.90 kg among underweight women; 11.50±2.82 kg among normal-weight women; 10.30±3.98 kg among overweight women; and 9.54±3.65 kg among obese women [6].

**Gestational weight gain according to INTERGROWTH-21st standard.** Three studies reported GWG according to the INTERGROWTH-21st standard [52, 57, 58]. One study reported that 27.5% of pregnant women gained gestational weight less than the third centile which is considered insufficient; 82.7% gained gestational weight less than the 50th centile; and 2.0% gained gestational weight above the 97th centile which is considered excess [58]. The other study reported that 26.8% of women with normal weight gained gestational weight less than the third centile, and none gained above the 97th centile [52].

**Gestational weight gain according to authors' classifications.** Akindele et al reported that 72.9% of women gained < 15 kg [44]; Onyiriuka reported that 42.9% of women gained < 10 kg [43]; Nkwabong reported that 75% of women gained <16 kg. [48]; Onwuka et al reported that 36.0% of women gained <10 kg [6], and Addo reported 14.8% of women gained ≤ 8.0 kg [39] (Table 5).

## Factors associated with gestational weight gain

Four studies reported factors associated with gaining weight according to IOM recommendations [2, 35, 52, 56]. These factors include pre-pregnancy weight [2, 35], having at least four ANC visits [2], engaging in physical activity [2, 35], income [2, 35], type of food consumption [2], knowledge about the importance of fruit [35], education [7, 35], type of food supplementation [52], and seasonality [56]. These factors are described below.

**Maternal pre-pregnancy weight.** According to two studies from Ethiopia [2, 35], women's early pregnancy BMI was associated with GWG. Asefa et al reported that overweight and

obese women were three times more likely to gain adequate gestational weight as compared to underweight women [2]. Similarly, Seifu reported that overweight and obese women were fourteen times more likely to have adequate GWG than those who were underweight [35].

**Table 4. Proportions of inadequate, adequate and excess gestational weight gain according to pre-pregnancy weight of the women in Sub-Saharan Africa, 2019.**

| Authors and year | Pre-pregnancy weight status of the women | Inadequate GWG n (%) | Adequate GWG n (%) | Excess GWG n (%) | Total n |
|---|---|---|---|---|---|
| Asefa F et al 2016 [2] | Underweight | 35 (89.7) | 3 (7.7) | 1 (2.6) | 39 |
| | Normal weight | 222 (75.0) | 71(24.0) | 3 (1.0) | 296 |
| | Overweight | 23 (38.3) | 31 (51.7) | 6 (10.0) | 60 |
| | Obese | 5(31.2) | 10 (62.5) | 1(6.3) | 16 |
| | Total | 285(69.3) | 115 (28.0) | 11(2.7) | 411 |
| Wanyama R et al 2016 [9] | Underweight | 20 (71.4) | 8 (28.6) | 0 (0.0) | 28 |
| | Normal weight | 98 (68.5) | 43 (30.1) | 2 (1.4) | 143 |
| | Overweight | 2 (9.5) | 15 (71.4) | 4 (19.1) | 21 |
| | Obese | 0(0.0) | 0(0.0) | 0(0.0) | 0 |
| | Total | 120 (62.5) | 66 (34.4) | 6 (3.1) | 192 |
| Wrottesley SV et al 2017 [54] | Underweight | 0(0.0) | 0(0.0) | 0(0.0) | 0 |
| | Normal weight | 54 (29.7) | 54 (29.7) | 74 (40.6) | 182 |
| | Overweight | 38 (20.0) | 32 (16.8) | 120 (63.2) | 190 |
| | Obese | 36 (21.7) | 27 (16.3) | 103 (62.0) | 166 |
| | Total | 128 (23.8) | 113 (21.0) | 297 (55.2) | 538 |
| Chithambo SET et al 2017 [56] | Underweight | 16 (88.9) | 2 (11.1) | 0(0.0) | 18 |
| | Normal weight | 163 (81.1) | 38 (18.9) | 0(0.0) | 201 |
| | Overweight | 27 (71.1) | 11 (28.9) | 0(0.0) | 38 |
| | Obese | 0(0.0) | 0(0.0) | 0(0.0) | 0 |
| | Total | 206 (80.2) | 51 (19.8) | 0(0.0) | 257 |
| Esimai OA et al 2017 [7] | Underweight | 46 (97.9) | 1 (2.1) | 0(0.0) | 47 |
| | Normal weight | 479 (99.4) | 2 (0.4) | 1 (0.2) | 482 |
| | Overweight | 45(97.8) | 1 (2.2) | 0(0.0) | 46 |
| | Obese | 0(0) | 14 (93.3) | 1 (6.7) | 15 |
| | Total | 570 (96.6) | 18 (3.1) | 2 (0.3) | 590 |
| Iyoke CA et al 2013 [42] | Underweight | NA* | NA* | NA* | NA* |
| | Normal weight | 109 (33.6) | 126 (38.9) | 89 (27.5) | 324 |
| | Overweight | NA* | NA* | NA* | NA* |
| | Obese | 12 (3.7) | 274 (84.6) | 38 (11.7) | 324 |
| | Total | 121 (18.7) | 400 (61.7) | 127 (19.6) | 648 |
| Seifu B 2017 [35] | Underweight | 84 (78.5) | 20 (18.7) | 3 (2.8) | 107 |
| | Normal weight | 268 (72.2) | 94 (25.4) | 9 (2.4) | 371 |
| | Overweight | 15 (23.1) | 44 (67.7) | 6(9.2) | 65 |
| | Obese | 2 (33.3) | 2 (33.3) | 2 (33.3) | 6 |
| | Total | 369 (67.2) | 160 (29.1) | 20 (3.7) | 549 |
| Agbayizah ED 2017 [53] | Underweight | 2 (66.7) | 1 (33.3) | 0 (0.0) | 3 |
| | Normal weight | 52 (31.7) | 62 (37.8) | 50 (30.5) | 164 |
| | Overweight | 9 (7.6) | 25 (21.0) | 85 (71.4) | 119 |
| | Obese | 10 (27.8) | 6 (16.7) | 20 (55.5) | 36 |
| | Total | 73 (22.7) | 94 (29.2) | 155 (48.1) | 322 |

NA*-Not applicable- because the authors (Iyoke et al) intended to compare GWG among normal weight and obese women, and they intentionally excluded underweight and overweight women

**Table 5. Proportions gestational weight gain in sub-Saharan Africa according to authors' classification, 2019.**

| Authors and year | Sample size | GWG classifications in kilogram | N (%) |
|---|---|---|---|
| Akindele et al 2017 [44] | 240 | <15 | 175 (72.9) |
| | | ≥15 | 65 (27.1) |
| Onyiriuka 2006 [43] | 408 | <10 | 175 (42.9) |
| | | 10 to 13 | 95 (23.3) |
| | | 13.1 to 15 | 129 (31.6) |
| | | ≥15 | 9 (2.2) |
| Elie N et al 2015 [48] | 232 | <16 | 174 (75.0) |
| | | ≥16 | 58 (25.0) |
| Onwuka et al 2017 [6] | 200 | <10 | 72 (36.0) |
| | | 10 to15 | 107 (53.5) |
| | | ≥15 | 21 (10.5) |
| Addo VN 2010 [39] | 1755 | ≤8 | 259 (14.8) |
| | | 8.1 to 16 | 1385 (78.1) |
| | | ≥16 | 111 (6.3) |
| Agbota G et al 2019 [40] | 253 | <7 kg | 65 (25.7) |
| | | 7 to 12 kg | 132 (52.2) |
| | | >>12 kg | 56 (22.1) |
| Tela FG et al 2019 [50] | 309 | <16 kg | 276 (89.3) |
| | | ≥16 kg | 33 (10.7) |

**Food consumption and physical activity.** Mothers' knowledge of the inclusion of fruits as a main food type during pregnancy was associated with gaining adequate gestational weight [35]. The women who ate fruit, vegetables, and meat at least once a week were more likely to gain adequate gestational weight compared with their counterparts [2]. According to Adu-Afarwuah et al, the percentage of women with adequate GWG was significantly higher in the group of women who received lipid-based nutrition supplementation than in a group who received multiple micronutrients and iron and folic acid supplementation [52]

Asefa et al reported that undertaking physical activity at least once a week for no less than 30 minutes was associated with higher likelihood of gaining adequate gestational weight [2]. Saifu also reported that engaging in physical activity up-to six hours a week was associated with gaining adequate gestational weight [35].

**Income, occupation, and social class.** One study reported that having a monthly family income of > $US100 was associated with gaining adequate gestational weight, while another study reported monthly income >$US150 as a factor associated with adequate GWG [2, 35]. According to Onwuka et al, women from a higher social class were more likely to gain weight of 10 to15 kg [6]. Being employed was reported as associated with gaining gestational weight of >7 kg [7].

**Maternal age and parity.** One study reported that being an adolescent (≤18 years of age) was associated with gaining gestational weight greater than 7 kg [7]. Another study reported that being younger than 35 years of age was associated with gaining gestational weight of 10 to 15 kg [6]. These two studies reported that being nulliparous was associated with gaining gestational weight of >7 kg [7] and 10 to 15 kg [6]. However, these associations are crude associations (not adjusted for confounders). According to Ouédraogo et al one increase in the number of pregnancies that a woman had was associated with increased odds of GWG below the 50[th] centile (OR, 1.11, 95% CI: 1.03, 1.20) [58]

**ANC visits.** Attending ANC four or more times was associated with gaining adequate gestational weight [2]. In addition, another study identified that having had regular ANC visits was associated with gaining gestational weight of 10 to 15 kg [6].

### Effect of GWG on birth outcomes

**Low birthweight.** An association between GWG and LBW was reported in some studies from SSA [42, 49, 51, 61].

Nemomsa et al reported that 17.5% of women who gained inadequate gestational weight gave birth to LBW babies, while 1.7% of women who gained adequate gestational weight gave birth to LBW babies[49]. Gondwe et al also reported that 15.6% of women who gained inadequate gestational weight gave birth to LBW babies; 7.6% of women who gained adequate gestational weight gave birth to LBW babies [28]; and none of the women who gained excess gestational weight in Nemomsa et al [49] and Gondwe et al [28] gave birth to LBW babies. In another study, 8.6% of women who gained inadequate weight, 11.5% of women who gained adequate weight, and 6.9% of women who gained excess gestational weight gave birth to LBW babies [45]. In another study, the proportion of LBW was 9.7% and 2.3% among women who gained < 10 kg and 10 to 15 kg, respectively [6].

**Macrosomia.** Seven studies reported an association between GWG and macrosomia [6, 43–47, 50]. Of these, five studies defined macrosomia as birth weight of ≥ 4 kg [6, 43–45, 50], while two studies did not clearly show how they defined macrosomia [46, 47]. Of the total seven studies, three studies classified GWG according to IOM [45–47], while the remaining four studies classified GWG according to their authors own classification [6, 43, 44, 50]. The percentage of a baby born with macrosomia was 30.9% [46], 11.0% [45] and 9.3% [47] among women who gained excessive gestational weight, while it was 3.9% [47] and 3.2% [45] among women who gained inadequate gestational weight. In other studies, the percentage was 83.1% [44], 66.7% [43], and 38.1% [6] among women who gained >15 kg, while it was 2.8% [6] and 20% [43] among women who gained <10 kg. Tela et al reported that 54.5% of women who gained ≥16 kg gave birth to a baby born with macrosomia while 16% of women who gained <16 kg gave birth to a baby born with macrosomia [50]. There was no statistically significant difference regarding giving birth to a baby born with macrosomia among women who gained adequate and excess gestational weight (RR, 0.68; 95% CI: 0.38, 1.50, P = 0.20), but this was based on two studies only (Fig 2).

**Caesarean section and episiotomy.** The percentage of caesarean section in two studies was 17% [47] and 26% [45] among women who gained inadequate gestational weight. The percentage was 10% [46], 16.7% [47], and 37% [45] among women who gained adequate weight; and 17.8% [47], 27.3% [46] and 50.3% [45] among women who gained excess gestational weight. According to Halle-Ekane *et al*, the percentages of episiotomy were 13.2%, 8.8% and 7.5% among women who gained inadequate, adequate and excess gestational weigh, respectively [47].

**Pre-eclampsia.** Pre-eclampsia was reported among 3.1% [47] and 7.5% [45] of women who gained inadequate gestational weight; 1.8% [47] and 6.4% [46] among those who gained adequate gestational weight; 15% [47], 18.2% [46], and 12.4% [45] among women who gained excess gestational weight. However, no significant difference was observed among women

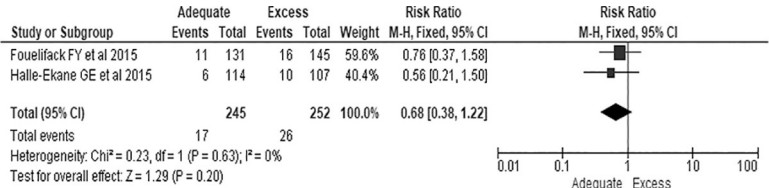

**Fig 2. The association of excess gestational weight gain and macrosomia in sub-Saharan Afric.**

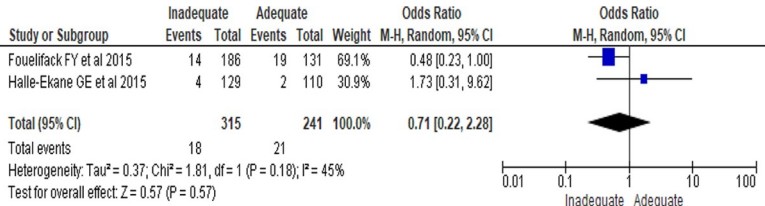

**Fig 3. The association of inadequate GWG and pre-eclampsia in sub-Saharan Africa.**

who gained inadequate compared with women who gained adequate gestational weight regarding predisposition to pre-eclampsia (RR, 0.71; 95% CI: 0.22, 2.28, P = 0.57) (Fig 3).

## Discussion

The percentage of inadequate GWG was >50% among nine of the 16 studies that classified GWG according IOM recommendations, and the percentage of inadequate GWG was particularly high among underweight women, ranging from 67% to 98%. High percentages of inadequate GWG were reported from low-income Sub-Saharan African countries (80% and 71.8% from Malawi [28, 56], 69.3% and 67.2% from Ethiopia [2, 35], 63% from Niger [58], 62.7% from Uganda [9], and 58.8% from Democratic Republic of Congo [59]) compared to middle-income countries (28% and 36.9% from Cameroon [45, 47], 15.7% and 24% from South Africa [54, 55]). Unlike in other high-income [20, 24, 62, 63] and middle-income [64] countries where many pregnant women experience excessive GWG, all of the studies from low-income Sub-Saharan countries [2, 9, 28, 35, 56, 58, 59] reported that more than 58% of pregnant women experienced inadequate GWG. This could be due to the inability of women to consume adequate food because of accessibility and affordability problems [65]. Pregnant women in low-income Sub-Saharan African countries suffer from a wide range of nutritional problems due to poverty, food insecurity and frequent infections [66]. Among seven of the sixteen studies, the percentages of women who gained excessive gestational weight were <10%. Five of these seven studies were from low-income countries (Ethiopia, Uganda and Malawi), and the percentage of excessive GWG among these studies were reported to be <6%. Seven studies where more than 10% the women gained excessive were from middle-income countries of Sub-Saharan Africa (Ghana, Nigeria, Cameroon, and South Africa). In South Africa, the percentage of women with excessive GWG was as high as 55%, which is even higher than for some studies from high-income countries such as Canada (49%) [67] and Australia (46%) [68]. The high percentage of excessive GWG may be explained by South Africa being an upper-middle income country [60], and 66% of participants in the South African study [54] were either overweight or obese. The finding of high levels of GWG in middle-income countries suggests the importance of low-income Sub-Saharan African countries designing strategies to prevent excessive GWG before it becomes a public health problem as these countries transition to middle-income countries.

While we pooled the percentages of the percentages of inadequate, adequate and excess GWG, substantial heterogeneity have been detected among studies, which may be explained by a number of factors. Firstly, the GWG classifications were inconsistent. Some authors classified GWG using the IOM recommendations while others used their own classifications. Secondly, studies were in different sub-Saharan African countries that had very different income levels, including upper middle- income, lower middle-income and lower-income countries. For example, a study from South Africa [54] reported that 55% of pregnant women gained excess gestational weight, whereas no women from Malawi [56] and <3% of pregnant women

from Ethiopia [2, 35] and Uganda [9] gained excess gestational weight. Thirdly, some studies were conducted in urban settings and in tertiary hospitals [42, 45], while others were conducted in semi-urban and rural [28, 51, 52, 56] settings in primary health care facilities. A study conducted in Nigeria in primary health care facilities [7] reported that 96.6% of pregnant women gained inadequate gestational weight. A study conducted in rural Malawi [56] showed that 80.2% of the pregnant women gained inadequate weight. By contrast the percentage of inadequate gestational weight gain was <30% among studies conducted in urban settings and tertiary hospitals [42, 45]. Finally, the difference in the pre-pregnancy weight of the participants may have affected the heterogeneity among studies. A study from South Africa [54] reported no underweight women; 66.2% of study participants were overweight and obese; and 55% of the participants gained excessive gestational weight. By contrast, studies from Malawi [56] and Uganda [9] had no obese women, and reported ≤ 3% of their participants gained excessive gestational weight.

This review identified that a number of factors that were associated with GWG including pre-pregnancy weight [2, 35], number and frequency of ANC visits [2], engaging in physical activity [2, 35], income [2, 35], type of food consumption [2], knowledge about the importance of fruit [35], education [7, 35], and type of food supplementation [52]. However, the inconsistent classification of the factors and poor control for confounding effects among the included studies made the findings of this review inconclusive.

The pre-pregnancy weight of women is associated with the amount of weight gained during pregnancy. Other studies have also reported that pre-pregnancy BMI is strongly associated with GWG [69–71]. This is because pre-gestational BMI is closely linked to maternal nutrition, lifestyle and socio-cultural factors, which could have an impact on the amount of GWG [72]. In this review, studies that have reported higher percentage of excessive GWG (for example, >30%) had a higher percentage of overweight and obese women (>46%) [45, 47, 53, 54]. Studies that have reported lower percentage of excessive GWG (<6%) had lower percentage of overweight and obese women (<15%) [2, 7, 9, 28, 35, 56]. These findings are supported by other studies that reveal a high BMI (overweight or obese) at the inception of pregnancy is associated with gaining weight above the IOM recommendations [70, 71]. In this review, the percentage of inadequate GWG ranged from 67% to 98% among underweight women [2, 7, 9, 35, 53, 56]. It may be difficult for underweight women to gain a sufficient amount of gestational weight, particularly if they tend to be underweight due to metabolic or food security factors [2]. Despite the association between pre-pregnancy weight and GWG, interventions on GWG managements took place mainly during pregnancy and focussed on reducing in GWG [73–76]. This implies that future interventions would need to focus on pre-pregnancy weight management strategies and its effectiveness. Weight management strategies should be inclusive by encouraging the reduction in GWG for women who are susceptible to excessive weight gain or encouraging weight gain for women who are susceptible to inadequate weight gain.

An association between GWG and birth weight has been reported by several studies [17, 77–79], and women who gain inadequate gestational weight are at an increased risk of having a baby with LBW or a pre-term birth [21, 80–82]. In the studies in this review, the majority of LBW babies were born to women who gained inadequate gestational weight. By contrast, a large percentage of women who gained excessive gestational weight (30%) gave birth to a baby born with macrosomia. However, the association between GWG and birth weight (LBW and macrosomia) were not statistically significant in the review that could be because of the small numbers of studies (only two studies) and small sample size (for example, only 6 of 196 [45], and 5 of 134 [47] pregnant women who gained inadequate gestational weight gave birth to a baby born with macrosomia). Several studies have reported that gaining GWG outside of IOM recommendations is associated with different adverse pregnancy outcomes such as caesarean

section [23, 69], episiotomy, low Apgar score at first and fifth minutes [83–85], antepartum haemorrhage, and pre-eclampsia [21]. However, given the inclusion of a limited number of studies in the meta-analyses, large differences in the settings among studies, and the inconsistent classification of GWG, these factors were not significantly associated with GWG outside of IOM recommendations in this analysis.

There were several issues relating to the quality of studies in the review. Firstly, the measurement of pre-pregnancy weight of the women was problematic, with four studies [9, 44, 45, 48] using self-reported pre-pregnancy weight. However, there is a typically a difference between self-reported weight and actual measured weight [86–88]. Women may be misclassified as gaining inadequate, adequate or excess based on self-reported pre-pregnancy weight [89]. Three studies [52, 54, 59] in this review used the weight of the women at 20 weeks of gestation and one study [56] used the weight of the women at 24 weeks of gestation as a proxy for pre-pregnancy weight. At this stage of pregnancy, there could be significant physiologic changes that may have resulted in weight gain, which may affect the measurement of GWG. In two studies [46, 58], it was unclear when or how the pre-pregnancy weight of the women was measured. The arbitrary classification of the outcome [6, 39, 40, 43, 44, 48, 50] and unclear classification of BMI and GWG were identified as a major quality issue in the review. Thirdly, there was poor control of confounding factors in many studies [6, 9, 39, 42, 43, 46–48]. These studies either did not control for confounding factors at all or did not include all necessary variables into the analysis (partially controlled). The identification of a number of quality issues in most of the included studies suggest the need for methodologically rigorous studies in sub-Saharan Africa to answer GWG related research questions including what factors affect GWG and the association between GWG and birth outcomes.

This review has a number of limitations. Firstly, the studies included in the review were highly heterogeneous and only two studies were eligible for the meta-analyses. Secondly, some of the included studies did not use standard GWG classifications. Thirdly, the pre-pregnancy weight of the women was assessed using different methods and at different stages (for example, pre-pregnancy or at 20 weeks). Fourthly, factors associated with GWG were classified inconsistently across studies. Finally, confounding factors were poorly controlled in most of the included studies.

## Conclusion

The percentage of inadequate GWG was as high as 80% in low-income Sub-Saharan countries while it was as low as 15% in upper middle-income Sub-Saharan African countries. In all studies from low-income Sub-Saharan countries, the percentage of inadequate GWG was greater than 58%. The percentage of inadequate GWG ranged from 67% to 98% among underweight women. Studies with a higher percentage of women with excessive GWG had a higher percentage of women who were overweight or obese, and those with a lower percentage of women with a lower percentage of excessive GWG had a lower percentage of women with high BMI (overweight or obese). Future interventions would need to give attention to design effective pre-pregnancy weight management strategies. Sub-Saharan African countries may need to develop regional GWG guidelines.

## Supporting information

**S1 Table. PRISMA-P 2015 checklist.**
(DOCX)

**S2 Table. Search strategies with corresponding database and numbers of articles accessed.**
(DOCX)

**S3 Table. Summary result of meta-analyses (Effect of inadequate GWG on Birth outcome).**
(DOCX)

**S4 Table. Summary result of meta-analyses (Effect of excessive GWG on Birth outcome).**
(DOCX)

**S1 Fig. Proportion of inadequate gestational weight gain in sub-Saharan Africa.**
(TIF)

**S2 Fig. Proportion of adequate gestational weight gain in sub-Saharan Africa.**
(TIF)

**S3 Fig. Proportion of excessive gestational weight gain in sub-Saharan Africa.**
(TIF)

**S4 Fig. Proportions inadequate gestational weight among underweight, normal weight, overweight and obese women in sub-Saharan Africa.**
(TIF)

**S5 Fig. Proportions adequate gestational weight among underweight, normal weight, overweight and obese women in sub-Saharan Africa.**
(TIF)

**S6 Fig. Proportions excessive gestational weight among underweight, normal weight, overweight and obese women in sub-Saharan Africa.**
(TIF)

## Acknowledgments

We acknowledge Beklecho Geleta's contribution in critically appraising papers.

## Author Contributions

**Conceptualization:** Fekede Asefa, Allison Cummins, Yadeta Dessie, Andrew Hayen, Maralyn Foureur.

**Data curation:** Fekede Asefa.

**Formal analysis:** Fekede Asefa.

**Methodology:** Fekede Asefa.

**Project administration:** Fekede Asefa.

**Supervision:** Allison Cummins, Yadeta Dessie, Andrew Hayen, Maralyn Foureur.

**Writing – original draft:** Fekede Asefa.

**Writing – review & editing:** Fekede Asefa, Allison Cummins, Yadeta Dessie, Andrew Hayen, Maralyn Foureur.

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
