## [Decision Letter · Decision Letter 0]

10 Dec 2019

PONE-D-19-28449

Gestational Weight Gain and its Effect on Birth Outcomes in sub-Saharan Africa: Systematic Review and Meta-analysis

PLOS ONE

Dear Mr Kumsa,

Thank you for submitting your manuscript to PLOS ONE. After careful consideration, we feel that it has merit but does not fully meet PLOS ONE’s publication criteria as it currently stands. Therefore, we invite you to submit a revised version of the manuscript that addresses the points raised during the review process.

ACADEMIC EDITOR:  The paper makes an important contribution to the literature by providing a comprehensive review of the effects of gestational weight gain on birth outcomes in the Sub-Saharan region. However, the description of methods and discussion of results needs improvement as pointed by the reviewers. Some the minor revisions may or may not be accepted by the authors, if an adequate counter-argument is presented. 

We would appreciate receiving your revised manuscript by Jan 24 2020 11:59PM. To enhance the reproducibility of your results, we recommend that if applicable you deposit your laboratory protocols in protocols.io, where a protocol can be assigned its own identifier (DOI) such that it can be cited independently in the future. For instructions see: http://journals.plos.org/plosone/s/submission-guidelines#loc-laboratory-protocols

We look forward to receiving your revised manuscript.

Kind regards,

Abraham Salinas-Miranda

Academic Editor

PLOS ONE

2. Please ensure that your search strategy includes studies that have been published in the past 12 months. If this is not appropriate please justify the reasons within the text.

Additional Editor Comments (if provided):

The paper makes an important contribution to the literature by providing a comprehensive review of the effects of gestational weight gain on birth outcomes in the Sub-Saharan region. However, the description of methods and discussion of results still needs improvement as pointed by the reviewers. Some the minor revisions may or may not be accepted by the authors, if an adequate counter-argument is presented.

Reviewers' comments:

Reviewer's Responses to Questions

**Comments to the Author**

1. Is the manuscript technically sound, and do the data support the conclusions?

Reviewer #1: No

Reviewer #2: Yes

Reviewer #3: Yes

2. Has the statistical analysis been performed appropriately and rigorously? 

Reviewer #1: No

Reviewer #2: Yes

Reviewer #3: Yes

3. Have the authors made all data underlying the findings in their manuscript fully available?

Reviewer #1: No

Reviewer #2: Yes

Reviewer #3: Yes

4. Is the manuscript presented in an intelligible fashion and written in standard English?

Reviewer #1: Yes

Reviewer #2: Yes

Reviewer #3: Yes

5. Review Comments to the Author

Reviewer #1: 1. Follow the Socioecological and the biopsychosocial model to address each one of the risk and protective factors affecting GWG and its effect on birth outcomes. Include them in the search terms. Evaluate the relationship found between these factors using statistics.

2. Consider cultural competency and variables specific for persons living in Sub-Saharan Africa, when evaluating articles regarding validity and reliability.

3. Evaluate the Obesity Paradox regarding GWG and birth outcomes.

4. Review statement: "recommendations on maternal weight management are inconsistent across

countries..."

Note: Guidelines regarding appropriate levels of weight gain in pregnancy have been promoted worldwide. A variety of guidelines about the GWG exist. About half of the GWG guidelines are similar to the 2009 American Institutes of Medicine (IOM) and 73% of the EIRs are similar to the 2006 IOM. Reference: N, Haley S, Chow K, McDonald SD. Comparison of national gestational weight gain guidelines and energy intake recommendations. Obes Rev 2013; 14:68.

5. Review statement: "To date, there are few systematic reviews and meta-analyses of research in sub-Saharan Africa

(SSA) on the weight of pregnant women [29-31]. None addressed how much weight is gained during pregnancy by women in this population, or the effect on birth outcomes."

Note: Include in your search "self-monitor weight throughout pregnancy," "self-care," "person-centered care."

6. Evaluate impact of Self-Care, Person-Centered Care, Person-Centered Medicine and People-Centered Public Health interventions on GWG and birth outcomes. How can these be implemented in Sub-Saharan Africa.

Reviewer #2: I reviewed the manuscript “Gestational Weight Gain and its Effect on Birth Outcomes in sub-Saharan Africa: Systematic Review and Meta-analysis”. The author(s) of this paper sought to present a review that determines the distribution of gestational weight gain (GWG) across African countries and its association with birth outcomes. Overall, I think this paper is well-written, and merits consideration for publication.

Some of the papers main strengths include a very clear presentation of findings of the studies considered in the review as well as meticulously critiquing each of the papers, identifying their merits and demerits. I, however, have some minor comments about issues the authors can clarify or address. Some of my concerns are outlined as below:

Minor comments:

Study selection procedure

I strongly suggest that the authors remove the names from the body of the article. Stating that “the authors screened the studies …” should suffice. The authors have clearly spelled out how they each contributed to the article at the end of it. Additionally, I don’t really think disagreements between authors and how they were resolved have any essence in the article. Please remove it.

Data extraction process

Again, remove the names of specific authors. I think all the authors bear responsibility for the merits and the demerits of the methods and procedures used in this study.

Study characteristics

I suggest the authors start a new sentence after Congo before proceeding to present the categories of the countries based on the income status. As it looks now, it reads as if the countries categorized based on income status are in addition to those listed by names. I would suggest starting the new sentence with “Based on the income status, these countries can be grouped into …”

Effect of GWG on birth outcomes

Low birthweight

While the results of Nenomsa et al. [39] show quite clear relationship between GWG and birth weight, Fouelifack et al. [41] seem to present quite a different picture. Were there any reasons presented in the study that could have accounted for a greater proportion of women who gained adequate GWG giving birth to LBW babies than those who did not gain adequate GWG?

Discussion

On page 23, change this sentence in the second paragraph:

“Firstly, the measurement of pre-pregnancy weight of the women was problematic, with four studies [9, 41, 52, 54] using self-reported pre-pregnancy weight. However, because there is a typically a difference between self-reported weight and actual measured weight [82-84].”

“Women tend to under-report their pre-pregnancy weight [85, 86], and…”

Substantively, I would suggest caution with attributing studies [85] and [86] to a study in sub-Saharan Africa. This is because I believe there is a social and cultural component that may render this statement not necessarily true for African women (there is quite a lot I could try to explain here but I will let you do that work on your end). Weight gain is not necessarily perceived as negative as it is in countries like the USA. I, therefore, suspect that women will be less likely inclined to deliberate understate their weight.

Summing it all up

I enjoyed reading the paper. It critiques that other studies very well, while it also does a significant amount of work outlining its own limitations. I suggest you consider the comments above and it will make an interesting read for your audience.

Reviewer #3: This is an interesting review. Please find my comments on how to improve it

1. ABSTRACT: better clarify why only 2 studies were included in the meta-analysis

2. Inclusion criteria. Add information on whether language was or not an exclusion criteria

3. Quality assessment of studies: I suggest to add details on the tool used

4. Data analysis. Please better clarify this sentence. Providing few examples “For studies that used arbitrary classifications, we used the authors’ own classifications.”

5. Terminology: I fell that “inadequate” GWG is confusing, and I suggest to modify this in “insufficient and “excessive”

6. Critical appraisal results: you can delete the following sentence which is already stated in methods “We used the Effective Public Health Practice Project Quality Assessment Tool for Quantitative Studies to assess the quality of the included studies

7. In methods will we good to further detail how you did analyzed factors associated with weight gain

8. Discussion: I suggest to shorten it down

9. Discussion: will be good to add a point in relation on how are the WHO recommendation o weight gain applicable (ie, based on evidence) to the African population

10. Discussion: add recommendations for policy makers and for researchers

6. PLOS authors have the option to publish the peer review history of their article (what does this mean?). If published, this will include your full peer review and any attached files.

Reviewer #1: No

Reviewer #2: No

Reviewer #3: Yes: Marzia Lazzerini

---

## [Author Response · Author response to Decision Letter 0]

24 Jan 2020

We have attached responses to reviewers comments

---

## [Decision Letter · Decision Letter 1]

31 Mar 2020

PONE-D-19-28449R1

Gestational Weight Gain and its Effect on Birth Outcomes in sub-Saharan Africa: Systematic Review and Meta-analysis

PLOS ONE

Dear Mr Kumsa,

Thank you for submitting your manuscript to PLOS ONE. After careful consideration, we feel that it has merit but does not fully meet PLOS ONE’s publication criteria as it currently stands. Therefore, we invite you to submit a revised version of the manuscript that addresses the points raised during the review process.

SPECIFIC ACADEMIC EDITOR COMMENTS: Please make the minor changes suggested by Reviewer #1.

We would appreciate receiving your revised manuscript by May 15 2020 11:59PM. To enhance the reproducibility of your results, we recommend that if applicable you deposit your laboratory protocols in protocols.io, where a protocol can be assigned its own identifier (DOI) such that it can be cited independently in the future. For instructions see: http://journals.plos.org/plosone/s/submission-guidelines#loc-laboratory-protocols

We look forward to receiving your revised manuscript.

Kind regards,

Frank T. Spradley

Academic Editor

PLOS ONE

Reviewers' comments:

Reviewer's Responses to Questions

**Comments to the Author**

1. If the authors have adequately addressed your comments raised in a previous round of review and you feel that this manuscript is now acceptable for publication, you may indicate that here to bypass the “Comments to the Author” section, enter your conflict of interest statement in the “Confidential to Editor” section, and submit your "Accept" recommendation.

Reviewer #1: All comments have been addressed

Reviewer #2: All comments have been addressed

2. Is the manuscript technically sound, and do the data support the conclusions?

Reviewer #1: Yes

Reviewer #2: Yes

3. Has the statistical analysis been performed appropriately and rigorously? 

Reviewer #1: Yes

Reviewer #2: Yes

4. Have the authors made all data underlying the findings in their manuscript fully available?

Reviewer #1: Yes

Reviewer #2: Yes

5. Is the manuscript presented in an intelligible fashion and written in standard English?

Reviewer #1: Yes

Reviewer #2: Yes

6. Review Comments to the Author

Reviewer #1: Very interesting and well written article.

Please review the following terms: "macrosomic baby" and consider writing instead "a baby born with macrosomia." Let us remember that the medical condition does not define the person.

Thank you

Reviewer #2: (No Response)

7. PLOS authors have the option to publish the peer review history of their article (what does this mean?). If published, this will include your full peer review and any attached files.

Reviewer #1: No

Reviewer #2: No

---

## [Author Response · Author response to Decision Letter 1]

1 Apr 2020

We have revised the manuscript as per the recommendation.

---

## [Editor Report · Decision Letter 2]

3 Apr 2020

Gestational Weight Gain and its Effect on Birth Outcomes in sub-Saharan Africa: Systematic Review and Meta-analysis

PONE-D-19-28449R2

Dear Dr. Kumsa,

We are pleased to inform you that your manuscript has been judged scientifically suitable for publication and will be formally accepted for publication once it complies with all outstanding technical requirements.

With kind regards,

Frank T. Spradley

Academic Editor

PLOS ONE

---

## [Editor Report · Acceptance letter]

7 Apr 2020

PONE-D-19-28449R2 

Gestational Weight Gain and its Effect on Birth Outcomes in sub-Saharan Africa: Systematic Review and Meta-analysis 

Dear Dr. Asefa:

I am pleased to inform you that your manuscript has been deemed suitable for publication in PLOS ONE. Congratulations! Your manuscript is now with our production department. 

With kind regards,

on behalf of

Dr. Frank T. Spradley 

Academic Editor

PLOS ONE